# Stem cell factor restrains endoplasmic reticulum stress-associated apoptosis through c-Kit receptor activation of JAK2/STAT3 axis in hippocampal neuronal cells

**Haiying Shen** [1] *, **Junjie Nie**[2], **Guangqing Li**[3], **Hongyan Tian**[4], **Jun Zhang**[4], **Xiaofeng Luo**[1], **Da Xu**[1], **Jie Sun**[1], **Dongfang Zhang**[1], **Hong Zhang**[1], **Guifang Zhao**[1], **Weiyao Wang**[1], **Zhonghua Zheng**[1], **Shuyan Yang**[1], **Yuji Jin**[5]

1 Department of Pathophysiology, School of Basic Medicine, Jilin Medical University, Jilin, Jilin Province, P. R. China, 2 Department of Nuclear Medicine, Jilin People's Hospital, Jilin, Jilin Province, P.R. China, 3 Department of Computer Application, School of Biomedical Engineering, Jilin Medical University, Jilin, Jilin Province, P.R. China, 4 Department of Histoembryology, School of Basic Medicine, Jilin Medical University, Jilin, Jilin Province, P.R. China, 5 Department of Medical Genetics, School of Basic Medicine, Jilin Medical University, Jilin, Jilin Province, P.R. China

* 3488936218@qq.com

**Data Availability Statement:** All relevant data are within the paper and its Supporting Information files

## Abstract

### Background

Alzheimer's disease (AD) is a common elderly disorder characterized by cognitive decline. Endoplasmic reticulum (ER) stress has been implicated in various neurodegenerative diseases, including AD. Stem cell factor (SCF) performs its biological functions by binding to and activating receptor tyrosine kinase c-Kit. We aimed to investigate the effects of SCF/c-Kit and JAK2/STAT3 on ER stress and apoptosis in AD.

### Methods

The study employed L-glutamic acid (L-Glu)-treated HT22 cells as sporadic AD cell model and APP/PS1 mice as an animal model of familiar AD. SCF, c-Kit inhibitor ISCK03 or JAK2/STAT3 inhibitor WP1066 was treated to verify the effects of SCF/c-Kit and JAK2/STAT3 on ER stress and apoptosis of L-Glu-exposed HT22 cells. Cell viability was assessed by MTT. BrdU detected cell proliferation. Flow cytometry measured cell apoptosis. The expression levels of ER stress markers GRP78, PERK, CHOP, and apoptosis protein caspase3 were determined by western blot. The effect on the mRNA of ER stress markers GRP78, PERK, CHOP and apoptotic caspase3 were quantified by RT-qPCR in primary cultured hippocampal neurons from APP/PS1 transgenic mice.

### Results

Administration of SCF significantly augmented the activity and proliferation of hippocampal neuronal cells, protecting cells against L-Glu induced ER stress-associated apoptosis.

**Funding:** This work was supported by grants from the "13th Five-Year" Science and Technology Project of Education Department of Jilin Province (JJKH20200456KJ).

Moreover, the addition of ISCK03 (c-Kit inhibitor) or WP1066 (JAK2/STAT3 inhibitor) reversed SCF effects on ER stress and apoptosis *in vitro*.

## Conclusion

We found that SCF inhibits L-Glu-induced ER stress-associated apoptosis via JAK2/STAT3 axis in HT22 hippocampal neuronal cells, as well as in primary hippocampal neurons from APP/PS1 mice, which provides valuable insights into the molecular mechanisms underlying the pathogenesis of AD and explores novel therapeutic targets for both sporadic and familial AD.

## Introduction

Alzheimer's disease (AD) is the most prevalent cause of dementia, and it has emerged as a burgeoning global health concern with profound impacts for individuals and society [1]. Cognitive impairment represents the primary symptom of AD patients, and progressive memory decline can be attributed to the degeneration of hippocampal neurons [2, 3]. The pathogenesis of AD is characterized by the misfolding and aggregation of proteins, which is accompanied by a prominent inflammatory component [4, 5]. Protein processing, modification and folding in the endoplasmic reticulum (ER) are strictly regulated processes that determine the function, fate and survival of cells [6]. ER stress, oxidative stress and inflammatory response constitute the main defense network, facilitating cells adapt to and survive stress conditions caused by biochemical, physiological and pathological stimuli [7]. The persistence of ER stress is believed to be the underlying mechanism driving numerous chronic diseases, which may induce abnormal inflammatory signals and promote cell death. The induction of ER stress and the concurrent accumulation of misfolded proteins in neurons contribute to neuronal dysfunction in neurodegenerative diseases (NDDs) [8]. Neuronal cells are particularly sensitive to abnormal protein folding, and ER stress has been implicated in various NDDs, including AD [9]. Therefore, it is imperative to investigate the mechanism of ER stress in AD for a comprehensive understanding of this pathological process.

Stem cell factor (SCF) is a hematopoietic growth factor, which can rapidly activate dormant stem cells and promote their growth, and also regulate the internal microenvironment of the body [10, 11]. SCF has been reported to exert neuroprotective effects by preventing neuronal apoptosis after acute spinal cord injury [12]. Studies have shown the plasma content of SCF is reduced in patients with NDDs, including AD. Conversely, the increased plasma concentration of SCF has shown potential for promoting AD treatment [13]. However, the precise mechanisms underlying SCF in AD remain unknown. In addition, SCF has been reported to perform its biological functions by binding to and activating receptor tyrosine kinase c-Kit (also referred to as stem cell factor receptor or CD117) [14, 15]. The tyrosine kinase receptor c-Kit and its ligand SCF play crucial roles in promoting cellular processes such as cell growth, survival, and proliferation [16]. Activation of c-Kit triggers autophosphorylation and initiates signal transduction [17]. Mutations in either c-Kit or SCF have implicated the involvement of c-Kit signaling in the cognitive function related to spatial learning within the brain's hippocampal domain [18]. Therefore, our study aims to investigate the role of SCF/c-Kit signaling in AD.

Janus kinase/signal transducer and activator of transcription (JAK/STAT) pathway is involved in the signaling cascades activated downstream of c-Kit, which leads to enhanced

phosphorylation of both JAKs and STATs [19]. The signaling pathway of JAK/STAT plays a pivotal role in the functioning of the cortex, hippocampus, and cerebellum, rendering it relevant to conditions associated with NDDs and other neuroinflammatory disorders [20]. Amyloid β (Aβ)-dependent inactivation of the JAK2/STAT3 axis in hippocampal neurons has been reported to cause cholinergic dysfunction through presynaptic and postsynaptic mechanisms, leading to AD-associated memory impairment [21]. Previous studies have demonstrated that the inhibition of JAK2/STAT3 pathway can enhance ER stress and exacerbate myocardial injury in experimental models of myocardial ischemia [22]. SERP1 restrains H9c2 apoptosis induced by hypoxia and reoxygenation through JAK2/STAT3 pathway-dependent repression of ER stress [23]. In addition, the activation of JAK2/STAT3 axis mediates neuroprotective effects and inhibits AD-related neurotoxicity [24]. These studies indicate a close association between the JAK2/STAT3 pathway and ER stress. However, it remains to be investigated whether SCF regulates JAK2/STAT3 axis to suppress ER stress in AD.

The mouse hippocampal neuronal cell line HT22 is increasingly being utilized in various studies on the pathogenesis of Alzheimer's disease, with particular emphasis on examining oxidative stress, mitochondrial dysfunction, ER stress, and apoptotic cell death. This cell line is widely acknowledged as a valuable vitro model for investigating the neurotoxicity induced by glutamate in relation to cognitive function and other neurodegenerative disorders [25]. The available evidence strongly supports the hypothesis that glutamate-induced cytotoxicity contributes to the pathogenesis of neurodegeneration and neuronal loss [26, 27]. The glutamate-triggered ER stress-mediated apoptosis in HT22 cell line has represented significant effectiveness in preclinical drug assessment [28, 29]. The double-transgenic amyloid precursor protein/presenilin 1 (APP/PS1) mice exhibit a chimeric mouse/human APP carrying the Swedish mutation (Mo/HuAPP695swe) and a mutant human PS1-dE9, both of which are implicated in familial AD [30, 31]. Transgenic mice exhibit cognitive decline and impaired memory starting at the age of 6 months, which also display significant neuropathological indicators resembling the early stages of familial AD.

Based on the above background, this study explored the effects of SCF/c-Kit and JAK2/STAT3 on ER stress and apoptosis in hippocampal neuronal cells, involving HT22 mouse hippocampal neuronal cells as sporadic AD cell model, as well as the primary cultured hippocampal neurons from APP/PS1 mice implicated in familial AD model. We found that SCF inhibits ER stress-associated apoptosis by activating the JAK2/STAT3 axis through the c-Kit receptor. Our research not only provides important clues for further understanding the molecular mechanisms underlying the occurrence and development of AD, but also paves the way for identifying potential novel therapeutic targets for both sporadic and familial AD.

## Materials and methods

### Cell culture and treatment

Mouse hippocampal neuron cell line HT22 was obtained from Sigma-Aldrich (SCC129, Sigma-Aldrich, USA). Cells were cultured in DMEM medium (D5796, Sigma-Aldrich, USA) containing 10% fetal bovine serum (FBS, 10099141, Gibco, USA) with 1% penicillin-streptomycin double antibody (SV30010, Beyotime Biotech, China) at 37°C, 5% $CO_2$ and saturated humidity. HT22 cells were induced by 5 mM L-Glutamic acid (L-Glutamate, L-Glu, G1251, Sigma-Aldrich, USA) to construct cell model and subsequently cultured for 72 h [29]. According to manufacturer's instructions, cells were treated with 25 ng/mL SCF (SRP3251, Sigma-Aldrich, USA) [32], 5 μM c-Kit inhibitor ISCK03 (I6410, Sigma-Aldrich, USA) or 5 μM JAK2/STAT3 inhibitor WP1066 (573097, Sigma-Aldrich, USA) [33], respectively.

## Cell viability assay

3-(4,5)-dimethylthiahiazo (-z-y1)-3,5-di- phenytetrazoliumromide (MTT, Sigma-Aldrich, USA) assay was performed to characterize cell viability. The cells at logarithmic growth stage were digested and counted. Cells were inoculated into 96-well plates ($1\times10^4$ cells/well, 100 μL per well). 0.5 mg/mL MTT (10 μL) were added to each well and cells were incubated at 37˚C for 4 h at 5% $CO_2$. Then the 96-well plate was retrieved, and the MTT-containing medium was aspirated prior to the addition of 100 μL of DMSO into each well. After shaking at room temperature for 15 min, the absorbance value at 490 nm was analyzed by Bio-Tek microplate analyzer (mb-530, Heales, China).

## BrdU incorporation and DAPI staining

The cell proliferation assay kit containing 5-bromo-2'-deoxyuridine (BrdU, BioVision, USA) was utilized to assess cellular proliferation following the manufacturer's instructions. In brief, cells were incubated in 1× BrdU medium for 2 h at 37˚C and washed with PBS twice. The cells were then fixed with fixing/denaturing solution. Then they were incubated with 1× BrdU detection antibody solution for 1 h. After washing, the cells were further incubated with anti-mouse HRP-linked antibody solution for another hour. The nuclei were counterstained with 4′,6-diamidino-2-phenylindole (DAPI, Sigma-Aldrich, USA) for 10 min. The fluorescence observations were conducted subsequent to the staining process using the confocal microscope (ZEISS, Oberkochen, Germany).

## Western blot analysis

The total protein of HT22 cells was extracted using RIPA lysis buffer (WB-0072, Dingguo, China). We followed instructions for BCA quantification kit to assess protein concentration. Equal quantities of protein were subjected to sodium dodecyl sulfate-polyacrylamide gel electrophoresis (SDS-PAGE) and transferred to polyvinylidene fluoride (PVDF) membranes (IPVH00010, Merck Millipore, USA). The membranes were probed with the primary antibodies against c-Kit (3074, 1:1000, CST), phospho (p)-c-Kit (3391, 1:1000, CST), JAK2 (ab108596, 1:5000, abcam), p-JAK2 (ab32101, 1:1000, abcam), STAT3 (ab68153, 1:1000, abcam), p-STAT3 (ab32143, 1:1000, abcam), GRP78 (GTX113340, 1:5000, GeneTex), CHOP (ab11419, 1:1000, abcam), PERK (phospho Thr982, GTX04533, 1:1000, GeneTex), caspase3 (GTX110543, 1:2000, GeneTex), SCF (GTX31379, 1:1000, GeneTex) and β-actin (ab8226, 1:1000, abcam) at 4˚C overnight. Then they were incubated with secondary antibodies. The enhanced chemiluminescence (ECL) chromogenic exposure was performed using Odyssey Infrared Imaging System (Li-COR Biosciences, USA) to assess protein bands, and β-actin was acted as an internal reference to detect expression levels. The blots were quantified by densitometric analysis using the ImageJ software.

## Apoptosis assay

The apoptotic status of cells was detected according to the instruction of Annexin V- fluorescein isothiocyante (FITC)/ propidium iodide (PI) cell apoptosis detection kit (KGA108, KeyGen, China). The kit utilizes annexin V conjugated with FITC to label phosphatidylserine sites on the membrane surface, along with PI to label the cellular DNA in necrotic cells with compromised cell membranes. The cells from each group were digested and collected by trypsin without EDTA. Cells were washed twice with PBS, followed by centrifugation at 1000 rpm for 5 min each time, approximately $2\times10^5$ cells were collected. Subsequently, 200 μL of binding buffer was added to suspend the cells. Then, the cells were mixed with 5 μL Annexin V-FITC

and 5 μL PI. The reaction time was 15 min at room temperature, away from light. Flow cytometry (CytoFLEX, Beckman, USA) was utilized for observation and detection within 1 h.

## Animals and primary hippocampal neurons culture

The *in vivo* study was approved by the Animal Care and Use Committee of Jilin Medical University (Approval Number: SCXK-20220003), and the informed written consent has been obtained from the committee, in accordance with ethical guidelines. The protocol described details regarding (1) methods of sacrifice, (2) methods of anesthesia and/or analgesia, and (3) efforts to alleviate suffering. Male APPswe/PS1dE9 (APP/PS1) double transgenic mice were come from Model Animal Research Center of Beijing HFK Bioscience Co., Ltd (Beijing, China). The transgenic mice have been shown to develop β-amyloid deposits in the cortex and hippocampus by 6 to 7 months of age [34]. Primary hippocampal neurons were isolated from the brain tissue of 6-month-old APP/PS1 mice. Isolation and culture of hippocampal neurons were performed as the previous study described [35]. All surgeries were conducted in an environment that was largely specific pathogen free. In brief, animals were anesthetized by placing in a container with isoflurane. After approximately 15 seconds of ceasing movement, remove the animal and confirm anesthesia by checking for the lack of a withdrawal reflex when pinching a toe. Decapitate mouse using a guillotine and disinfect the head with 70% ethanol. Rapidly dissect hippocampus from the rodent brain. The hippocampal cells were resuspended in Hibernate-A medium (A1247501, Gibco, USA) supplemented with 2% B27 supplements (17504044, Gibco, USA) and 0.5 mM L-glutamine (25030081, Gibco, USA). They were seeded on poly-L-lysine-coated (PLL, A3890401, Gibco, USA, 0.1 mg/ml) 12-well (1.0 × 106 cells/well) culture plates. The medium was replaced with Neurobasal medium (21103049, Gibco, USA) supplemented with 2% B27 supplements (17504044, Gibco, USA) and 0.5 mM L-glutamine (25030081, Gibco, USA) 24 h later. After 7 days, the cells were administrated with SCF (25 ng/mL), the c-Kit inhibitor ISCK03 (5 μM) and the JAK2/STAT3 inhibitor WP1066 (5 μM) for 24 h, and the experiments were carried out with Model group, SCF group, SCF+ISCK03 group and SCF+WP1066 group.

## Quantitative real-time PCR (qRT-PCR)

Total RNA was extracted by Trizol method and reverse transcribed into cDNAs by RevertAid Reverse Transcriptase (EP0441, Thermo, USA). PerfectStart Green qPCR SuperMix (AQ601-04, TransGen Biotech, China) was used and tested relative gene expression on ABI QuantStudio 1 system (Applied Biosystems, USA). GAPDH was served as an internal reference gene, the relative gene levels were calculated by the $2^{-\Delta\Delta Ct}$ method. The primer sequences were shown in list (5'-3'): GRP78 forward, AGCAGGACATCAAGTTCTTGCC; GRP78 reverse, CTTGTCGCTGGGCATCATTG; CHOP forward, CTCATCCCCAGGAAACGAAGAG; CHOP reverse, CCGTTTCCTAGTTCTTCCTTGCTC; PERK forward, ATGGACGAATCGCTGCACTG; PERK reverse, GAAGTTTTGTGGGTGCCCTCT;

Caspase3 forward, TGTCATCTCGCTCTGGTACG; Caspase3 reverse, GTTCAACAGGCCCATTTGTC; GAPDH forward, AGCCCAAGATGCCCTTCAGT; GAPDH reverse, CCGTGTTCCTACCCCCAATG.

## Statistical analysis

Graphpad Prism 8.0 software was used for statistical analysis. Mean ± standard deviation (SD) was used for measurement data, and unpaired T test was used between two groups conforming to normal distribution. Experimental results were analyzed by one-way ANOVA for ranked

data followed by Tukey's multiple comparison test. P < 0.05 indicated statistically significant difference.

## Results

### SCF protects HT22 cells against L-Glu-induced ER stress-associated apoptotic cell death

To investigate the role of SCF on neuronal viability, HT22 cells were treated with L-Glu or SCF respectively. Cell viability was assessed using MTT assay, while BrdU incorporation was employed to detect cell proliferation. Flow cytometry analysis was conducted to quantify cell apoptosis. In comparison with control group, L-Glu exposure resulted in a reduction in cellular activity and proliferation, accompanied by an increase in apoptotic cell death. However, after administration of SCF, there was a significant enhancement in cell activity and proliferation, while apoptosis was notably attenuated (Fig 1A–1C). Next, to further analyze the impact of SCF on ER stress in HT22 cells, the expression levels of ER stress markers were examined. Additionally, CHOP serves as the principal transcription factor that connects ER stress to apoptotic cell death, and it was observed to be overexpressed in the brains of patients with Alzheimer's disease, alongside downstream effectors like apoptotic caspases [4, 36]. Given the involvement of CHOP and caspase3 in ER stress-induced apoptosis, the expression of these two proteins was also investigated. As shown in Fig 1D, upon L-Glu stimulation, SCF expression was suppressed, while the expressions of ER stress markers GRP78, PERK, CHOP, and apoptosis protein caspase3 were observed to be elevated. In contrast, treatment with SCF markedly reduced the expression of ER stress markers and apoptotic proteins (Fig 1D). These results suggested that the administration of SCF significantly augments the activity and proliferation of HT22 cells, protecting cells against L-Glu-induced ER stress-associated apoptosis.

### SCF modulates ER stress-associated apoptosis of HT22 cells by regulating c-Kit phosphorylation

In order to examine the effect of inhibiting SCF-mediated c-Kit phosphorylation on HT22 cells, we next added SCF/c-Kit inhibitor ISCK03 to suppress SCF-mediated c-Kit phosphorylation. The compound ISCK03 functions as a tyrosine kinase inhibitor specific to SCF/c-Kit signaling pathway, effectively suppressing SCF-induced phosphorylation of c-Kit [37, 38]. The level of c-Kit phosphorylation was observed to decrease following L-Glu exposure, whereas the phosphorylation of c-Kit exhibited markable significant enhancement in response to SCF administration. However, the subsequent addition of ISCK03 resulted in the remarkable suppression of c-Kit phosphorylation (Fig 2A). Inhibition of c-Kit phosphorylation attenuated the cell viability and proliferation promoted by SCF treatment, while concurrently enhancing cellular apoptosis (Fig 2B–2D). The findings indicated the cytotoxicity caused by L-Glu was effectively ameliorated upon treatment with SCF, whereas this protective effect was counteracted upon inhibition of c-Kit phosphorylation. We further explored the impact of c-Kit on ER stress in HT22 cells. The expression levels of ER stress marker GRP78, PERK, CHOP, and apoptotic protein caspase3 were quantified. The findings showed that administration of SCF effectively suppressed the expression of ER stress markers and apoptotic caspase3; however, this beneficial effect was reversed by inhibiting the level of c-Kit phosphorylation (Fig 2E). The above results demonstrated that SCF modulated ER stress-associated apoptosis through regulation of c-Kit phosphorylation.

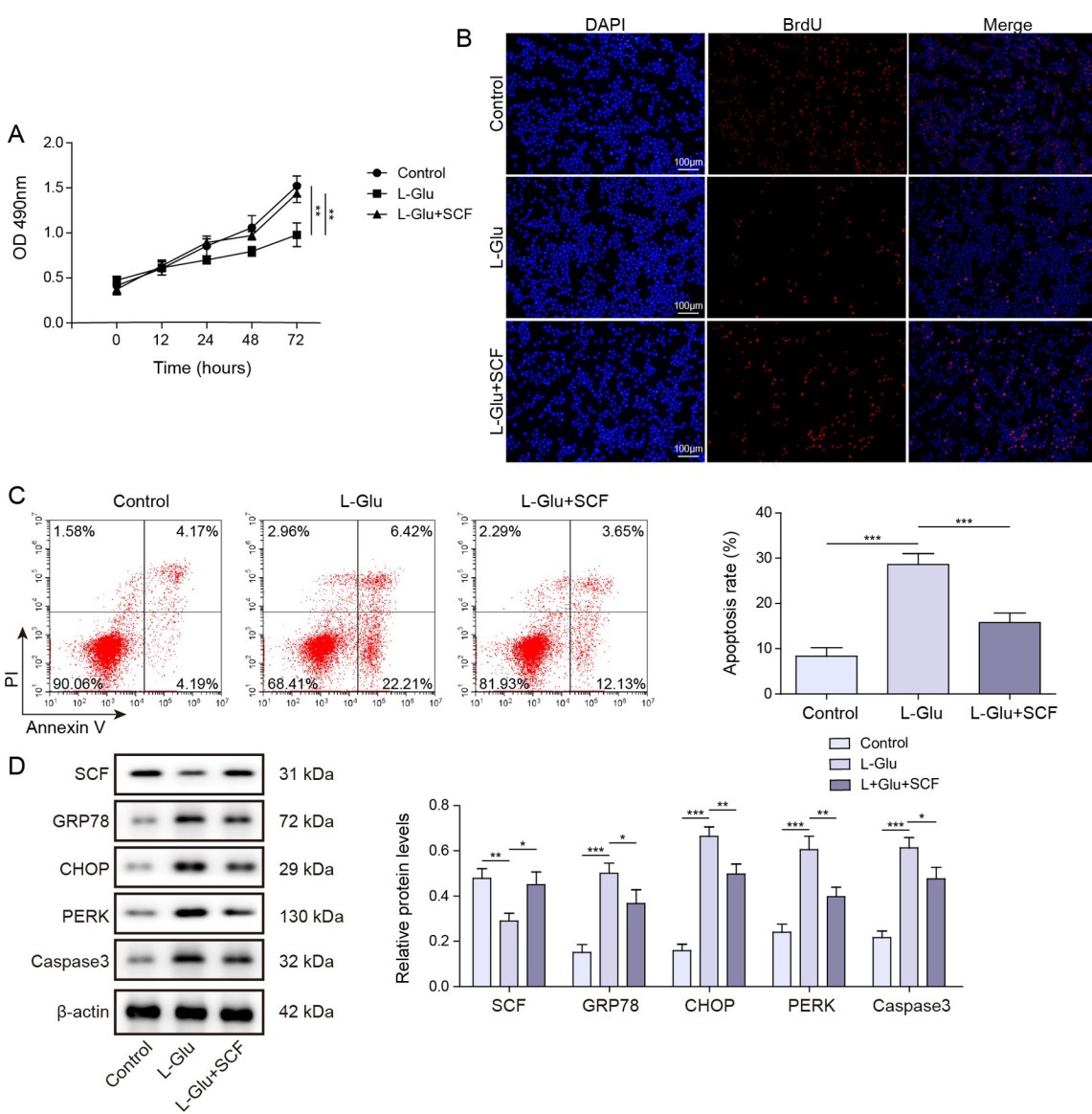

**Fig 1. SCF reduces ER stress-associated apoptosis of HT22 cells.** The effect of SCF was assessed by treating HT22 cells with either L-Glu or SCF respectively. (A) Cell viability was measured using an MTT assay after treatment with 5 mM L-Glu. Relative survival was expressed as mean values and statistical significance was calculated using three independent experimental replicates. ** P < 0.01. (B) BrdU detected cell proliferation. The data were representative of three independent experiments per group. (C) Flow cytometry measured cell apoptosis. Data were means ± SD, and statistical significance was calculated using three independent experimental replicates. *** P < 0.001. (D) SCF, ER stress markers GRP78, CHOP, PERK and apoptosis protein caspase3 expressions were detected by western blot. The data were expressed as mean ± SD, and statistical significance was calculated using triplicate experiments. * P < 0.05, ** P < 0.01, *** P < 0.001.

## c-Kit activates JAK2/STAT3 signaling pathway to repress ER stress-associated apoptosis of HT22 cells

To further elucidate the effect of c-Kit on the JAK2/STAT3 pathway, the phosphorylation levels of JAK2 and STAT3 were examined. We observed a remarkable attenuation in the phosphorylation of JAK2 and STAT3 upon L-Glu stimulation, whereas treatment with SCF significantly enhanced the levels of phosphorylated JAK2 and STAT3. Conversely, the addition of c-Kit inhibitor ISCK03 markedly suppressed the phosphorylation levels of JAK2 and

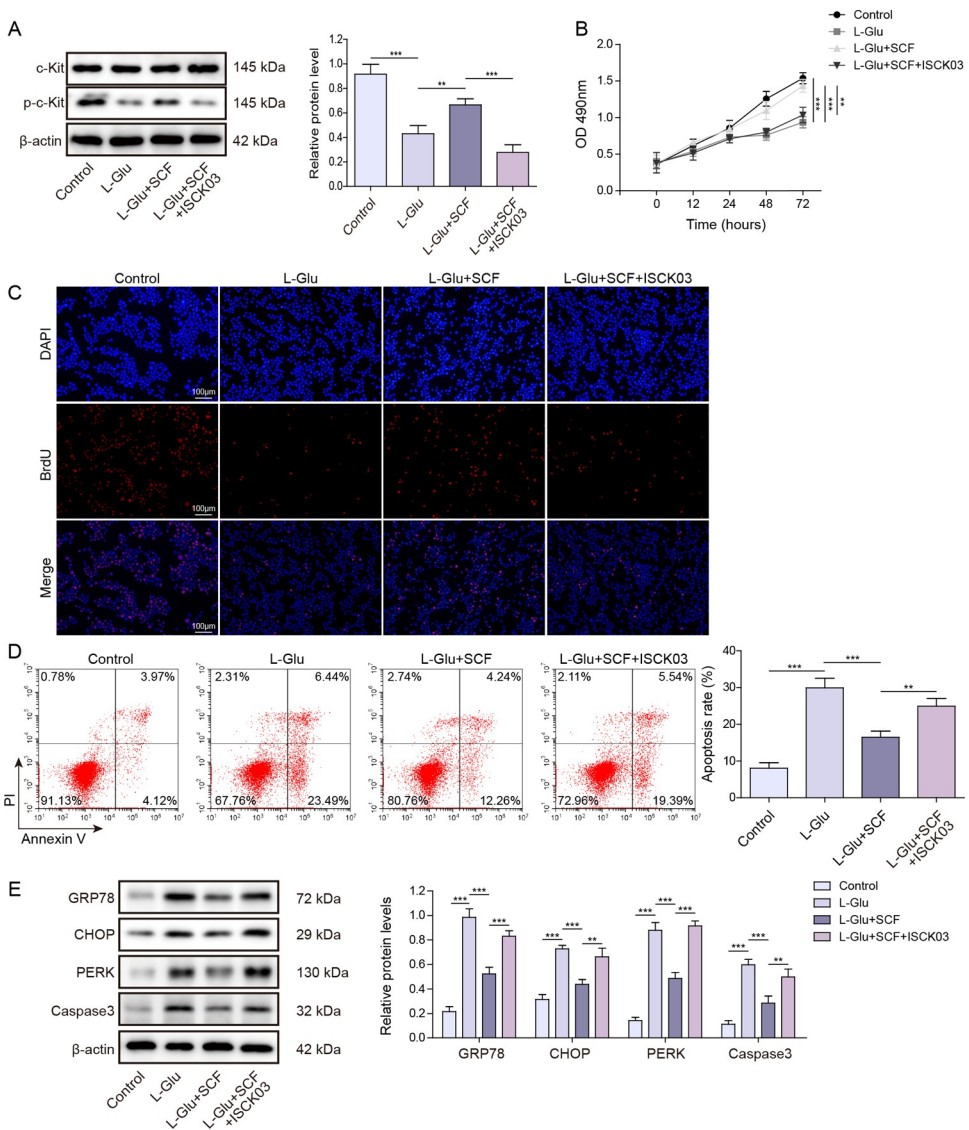

**Fig 2. SCF regulates the effects of c-Kit phosphorylation on ER stress-associated apoptosis of HT22 cells.** The specific SCF/c-Kit inhibitor ISCK03 was added to inhibit SCF-mediated c-Kit phosphorylation. (A) Western blot detected p-c-Kit expression in cells. Data were means ± SD, and statistical significance was calculated using three independent experimental replicates. ** $P < 0.01$, *** $P < 0.001$. (B) MTT was used to detect cell activity. Relative survival was expressed as mean values and statistical significance was calculated using three independent experimental replicates. ** $P < 0.01$, *** $P < 0.001$. (C) BrdU detected cell proliferation. The data were representative of three independent experiments per group. (D) Flow cytometry measured cell apoptosis. Data were means ± SD, and statistical significance was calculated using three independent experimental replicates. ** $P < 0.01$, *** $P < 0.001$. (E) Expression of ER stress markers GRP78, CHOP, PERK and apoptosis protein caspase3 were determined by western blot. The data were expressed as mean ± SD, and statistical significance was calculated using triplicate experiments. ** $P < 0.01$, *** $P < 0.001$.

STAT3 (Fig 3A). To further investigate the impact of JAK2/STAT3 pathway on ER stress and apoptosis, we employed WP1066, a small molecule inhibitor targeting the JAK2/STAT3 signaling cascade [39]. As shown in Fig 3B, WP1066 inhibited the phosphorylation levels of JAK2 and STAT3. A series of cell viability experiments indicated the addition of WP1066 counteracted the protective effects of SCF treatment on cellular activity and proliferation, as well as apoptosis inhibition in response to L-Glu toxicity (Fig 3C–3E). We further examined the

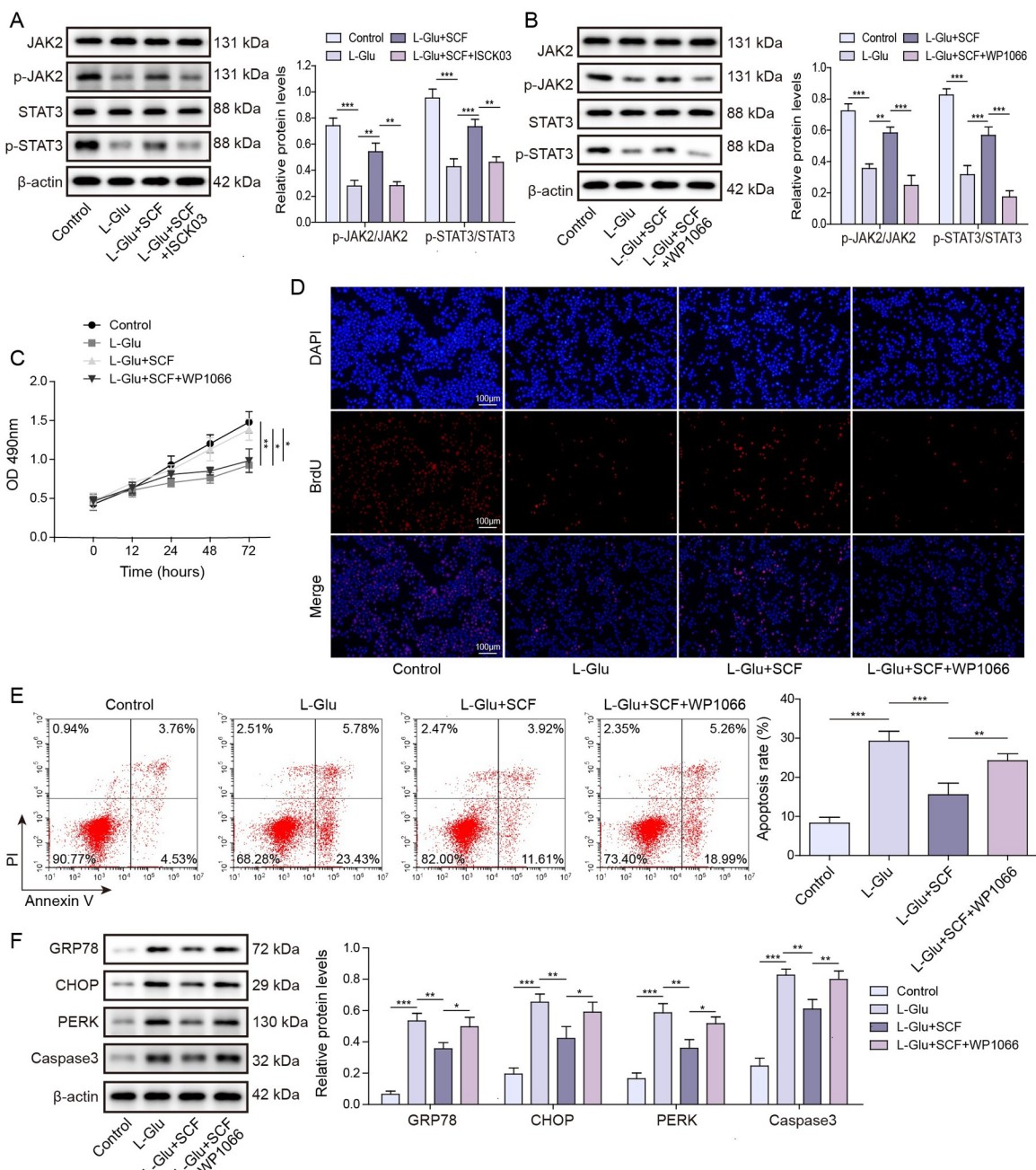

**Fig 3. c-Kit activates JAK2/STAT3 pathway to affect ER stress-associated apoptosis of HT22 cells.** c-Kit inhibitor ISCK03 or JAK2/STAT3 inhibitor WP1066 was administered to investigate the impact of SCF/c-Kit on the JAK2/STAT3 pathway. (A) Western blot detected JAK2 and STAT3 phosphorylation in cells, with the addition of the c-Kit inhibitor ISCK03 to the cells. Data were means ± SD, and statistical significance was calculated using three independent experimental replicates. ** $P < 0.01$, *** $P < 0.001$. (B) The phosphorylation of JAK2 and STAT3 in cells was detected by western blot analysis after the administration of JAK2/STAT3 inhibitor WP1066. Data were means ± SD, and statistical significance was calculated using three independent experimental replicates. ** $P < 0.01$, *** $P < 0.001$. (C) MTT was used to detect cell activity. Relative survival was expressed as mean values and statistical significance was calculated using three independent experimental replicates. * $P < 0.05$, ** $P < 0.01$. (D) BrdU detected cell proliferation. The data were representative of three independent experiments per group. (E) Flow cytometry evaluated cell apoptosis. Data were means ± SD, and statistical significance was calculated using three independent experimental replicates. ** $P < 0.01$, *** $P < 0.001$. (F) Expression of ER stress markers GRP78, CHOP, PERK and apoptosis protein caspase3 were evaluated by western blot. The data were expressed as mean ± SD, and statistical significance was calculated using three independent experimental replicates. * $P < 0.05$, ** $P < 0.01$, *** $P < 0.001$.

relevant changes in the expression of ER stress markers and apoptotic proteins induced by WP1066. The findings shown that in the L-Glu-injured cells, the addition of WP1066 notably enhanced the upregulation of ER stress markers, including GRP78, PERK, and CHOP, as well as apoptotic caspase3 when compared to treatment with SCF alone (Fig 3F). These results indicated that SCF/c-Kit played the role that activated the JAK2/STAT3 pathway in HT22 cells, and its inhibition of ER stress-induced apoptosis was mediated through c-Kit-induced activation of the JAK2/STAT3 signaling cascade.

## SCF ameliorates ER stress-associated apoptosis in primary hippocampal neurons from APP/PS1 mice

To further verify the protective effects of SCF and its underlying pathway, the primary hippocampal neurons from APP/PS1 transgenic mice were administrated with SCF, the SCF/c-Kit inhibitor ISCK03, or the JAK2/STAT3 inhibitor WP1066, respectively. The relative mRNA levels of ER stress markers GRP78, PERK, CHOP, and apoptosis protein caspase3 were investigated by RT-qPCR. As shown in Fig 4, compared to the Model group, the mRNA expressions of GRP78, CHOP, PERK and caspase3 were significantly decreased by treating with SCF, while this effect was suppressed due to the addition of SCK03 or WP1066. The results demonstrated that treatment with SCF effectively attenuate gene expression levels of ER stress-associated apoptosis in primary hippocampal neurons from APP/PS1 mice by activating the JAK2/STAT3 axis through the c-Kit receptor.

## Discussion

AD is a neurodegenerative disease characterized by abnormal accumulation of β-amyloid protein, neurofibrillary tangles and neuronal cell death [40, 41]. It has been reported that there is a feedback loop between neuroinflammation and ER stress, which is closely related to the neurodegenerative process of AD [42]. Previous studies have shown that Aβ-dependent inactivation of JAK2/STAT3 axis in hippocampal neurons causes cholinergic dysfunction through presynaptic and postsynaptic mechanisms, leading to AD-related memory impairment [22]. SCF is a hematopoietic growth factor (HGF) that promotes neuroprotection and supports neurogenesis in the brain [43]. However, the effects of SCF/c-Kit and JAK2/STAT3 on ER stress and apoptosis in sporadic and familial AD remain unclear. In this research, we employed HT22 hippocampal neuronal cells and primary hippocampal neurons from APP/PS1 mice to validate *in vitro* cell experiments that SCF inhibits ER stress and apoptosis through c-Kit receptor activation of JAK2/STAT3 axis.

ER is a crucial organelle involved in protein quality control and cell homeostasis [9]. The pathogenesis of AD is intricately associated with ER stress [44]. Aggressive accumulation of Aβ accelerates the formation of senile plaques and disrupts ER function in AD [45]. Perturbations in protein homeostasis caused by ER stress can lead to irregular accumulation of proteins [42]. ER stress-associated neuronal apoptosis is considered a pivotal factor in the progression and pathology of neurodegeneration in AD [46]. Activation of CHOP through ER stress is a key factor in the onset of AD pathological characteristics, as evidenced by the induction of oxidative stress, misfolded protein aggregation, and neuronal inflammation, ultimately leading to neuronal apoptosis [47]. It is imperative to identify potential targets and restore the equilibrium between apoptosis and anti-apoptosis to determine neuronal fate and develop prospective drugs for AD [48]. Previous study reported the expression of SCF and c-Kit was upregulated after spinal cord injury, and SCF can prevent the apoptosis of nerve cells after acute spinal cord injury [12]. Consistent with the report, we observed that the administration of SCF resulted in a significant mitigation of ER stress-induced apoptotic cell death, as

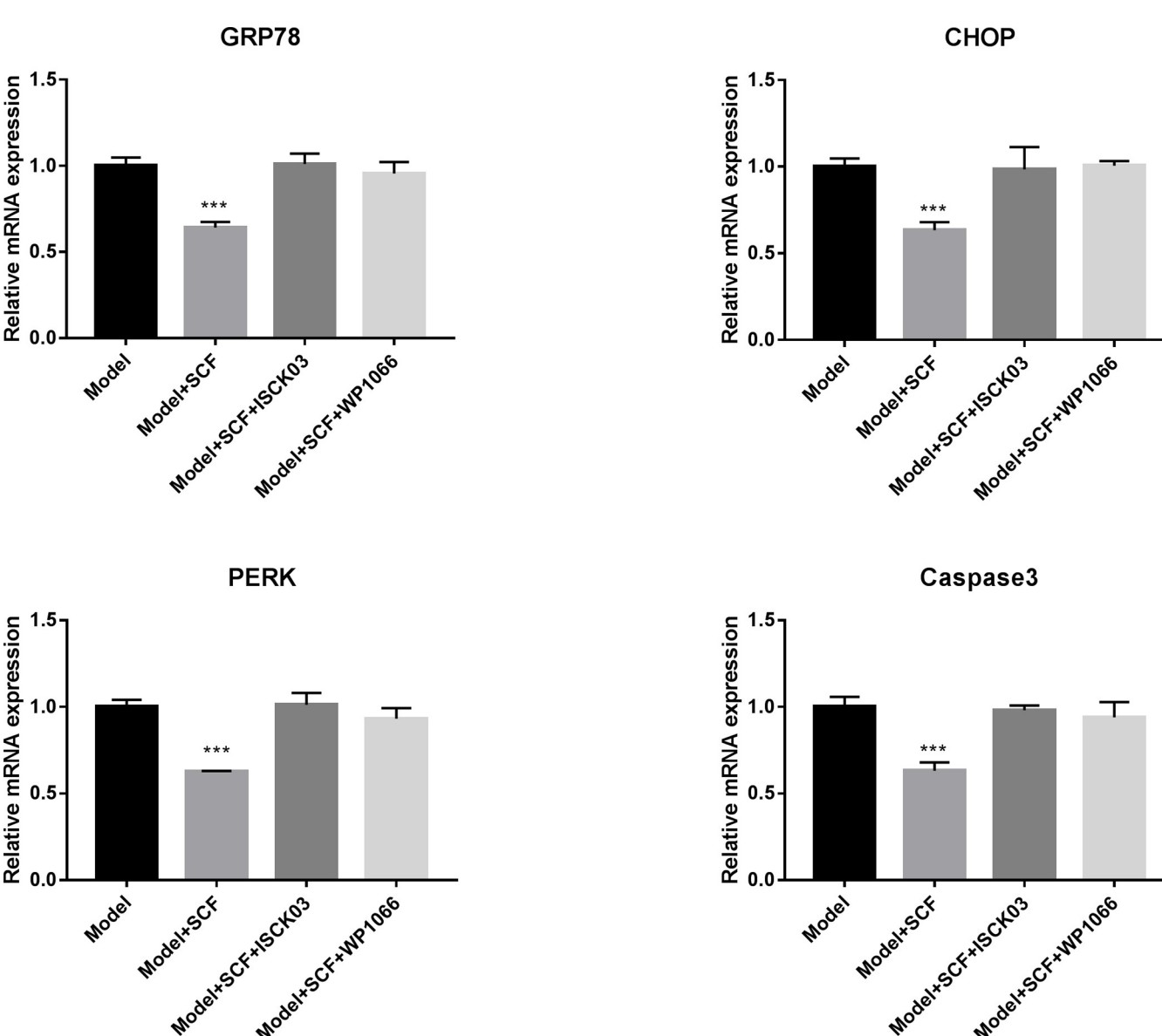

**Fig 4. SCF ameliorates ER stress and apoptosis in hippocampal neurons from APP/PS1 mice.** Relative mRNA expressions of ER stress markers GRP78, PERK, CHOP and apoptosis protein caspase3 were detected by RT-qPCR in primary cultured hippocampal neurons from APP/PS1 transgenic mice. Quantification of the mRNA levels of GRP78, PERK, CHOP and caspase3 in hippocampal neurons of APP/PS1 mice was detected by quantitative real-time PCR. Unpaired Student's t-test, and statistical significance was calculated using three independent experimental replicates. ***P<0.001.

evidenced in HT22 hippocampal neuronal cells and primary hippocampal neurons derived from APP/PS1 mice. After inhibiting c-Kit phosphorylation, ER stress-associated apoptosis in hippocampal neuronal cells was enhanced. Our study provides new evidence for the first time that the application of SCF can effectively alleviate cell death induced by ER stress in AD cell models, thereby restoring cellular homeostasis and preventing neuronal loss.

The JAK2/STAT3 pathway represents an α7 nAChR-mediated cholinergic anti-inflammatory mechanism in macrophages, microglia, and neurons [49]. The JAK2/STAT3 axis serves as the main mediator of humanin-induced neuroprotective activity [22]. It has been reported that humanin represses neuronal death by binding to specific receptors on the cell membrane

and triggering JAK2/STAT3 pro-survival pathway [50]. Long QH, et al.'s study found that low-dose Suan-Zao-Ren decoction can inhibit Aβ accumulation and neuroinflammation through JAK2/STAT3 pathway, improving cognitive function damage and neurodegeneration in APP/PS1 transgenic mice [51]. These researches suggest that activation of JAK2/STAT3 pathway inhibits neuronal death. In our study, the addition of JAK2/STAT3 inhibitor WP1066 resulted in an exacerbation of ER stress-induced apoptotic cell death in hippocampal neuronal cells, indicating that c-Kit activation of the JAK2/STAT3 signaling pathway suppresses apoptosis associated with ER stress. This effect was observed in both HT22 hippocampal neuronal cells and primary hippocampal neurons derived from APP/PS1 mice.

The double-transgenic APP/PS1 mice with mutations associated with familial AD manifest cognitive decline and impaired memory, resembling neuropathological indicators of familial AD [30, 31]. Our finding demonstrates that treatment of SCF effectively reduces the expression levels of genes involved in ER stress-induced apoptosis in primary hippocampal neurons derived from APP/PS1 mice. This reduction is achieved by activating the JAK2/STAT3 pathway through stimulation of the c-Kit receptor. In the present study, the limited sample size of transgenic mouse hippocampal neurons posed a significant obstacle in accurately determining protein expression levels, such as through the use of western blotting methods. To address this issue, our future studies will prioritize increasing the sample size to further elucidate the differences in corresponding protein expressions and potential mechanisms involved.

In conclusion, we investigated the effects of SCF/c-Kit and JAK2/STAT3 on ER stress-associated apoptotic cell death in HT22 hippocampal neuronal cells and primary hippocampal neurons derived from APP/PS1 mice. Our findings suggest that SCF plays a crucial role that significantly restrains ER stress-associated apoptotic cell death through c-Kit receptor activation of JAK2/STAT3 axis, which provides valuable insights into the molecular mechanisms underlying the pathogenesis of AD and explores novel therapeutic targets for both sporadic and familial AD.

## Supporting information

**S1 Raw data. Fig 1 total raw data.**
(RAR)

**S2 Raw data. Fig 2 total raw data.**
(RAR)

**S3 Raw data. Fig 3 total raw data.**
(RAR)

**S4 Raw data. Figs 1–3 WB images and experimental reports.**
(RAR)

**S5 Raw data. Fig 4 QPCR experimental reports.**
(RAR)

## Author Contributions

**Conceptualization:** Hongyan Tian.

**Data curation:** Junjie Nie, Jun Zhang, Zhonghua Zheng.

**Formal analysis:** Guifang Zhao.

**Funding acquisition:** Haiying Shen, Junjie Nie.

**Investigation:** Shuyan Yang.

**Methodology:** Guangqing Li, Hong Zhang, Weiyao Wang.

**Project administration:** Haiying Shen.

**Resources:** Da Xu, Jie Sun.

**Software:** Guangqing Li, Xiaofeng Luo.

**Validation:** Dongfang Zhang, Yuji Jin.

**Writing – original draft:** Haiying Shen, Junjie Nie.

**Writing – review & editing:** Haiying Shen.

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
