## [Decision Letter · Decision Letter 0]

9 Jul 2024

PONE-D-24-18303Stem cell factor restrains endoplasmic reticulum stress-associated apoptosis through c-kit receptor activation of JAK2/STAT3 axis in hippocampal neuronal cellsPLOS ONE

Dear Dr. SHEN,

Thank you for submitting your manuscript to PLOS ONE. After careful consideration, we feel that it has merit but does not fully meet PLOS ONE’s publication criteria as it currently stands. Therefore, we invite you to submit a revised version of the manuscript that addresses the points raised during the review process.

We look forward to receiving your revised manuscript.

Kind regards,

Abhishek Kumar Singh, Ph.D.

Academic Editor

PLOS ONE

Additionally, please provide the animal ethics approval number along with the animal ethics approval committee.

 [This work was supported by grants from the "13th Five-Year" Science and Technology Project of Education Department of Jilin Province (JJKH20200456KJ)].  

5. PLOS requires an ORCID iD for the corresponding author in Editorial Manager on papers submitted after December 6th, 2016. Please ensure that you have an ORCID iD and that it is validated in Editorial Manager. To do this, go to ‘Update my Information’ (in the upper left-hand corner of the main menu), and click on the Fetch/Validate link next to the ORCID field. This will take you to the ORCID site and allow you to create a new iD or authenticate a pre-existing iD in Editorial Manager. Please see the following video for instructions on linking an ORCID iD to your Editorial Manager account: https://www.youtube.com/watch?v=_xcclfuvtxQ.

Additional Editor Comments:

Please provide uncropped western blotting images as supplementary files. 

Reviewers' comments:

Reviewer's Responses to Questions

**Comments to the Author**

1. Is the manuscript technically sound, and do the data support the conclusions?

Reviewer #1: Yes

Reviewer #2: Partly

Reviewer #3: Yes

2. Has the statistical analysis been performed appropriately and rigorously? 

Reviewer #1: Yes

Reviewer #2: Yes

Reviewer #3: Yes

3. Have the authors made all data underlying the findings in their manuscript fully available?

Reviewer #1: Yes

Reviewer #2: No

Reviewer #3: Yes

4. Is the manuscript presented in an intelligible fashion and written in standard English?

Reviewer #1: Yes

Reviewer #2: Yes

Reviewer #3: Yes

5. Review Comments to the Author

Reviewer #1: The article is written and the hypothesis is work well. Authors indicated that the findings provides valuable insights into the molecular mechanisms underlying the pathogenesis of AD and explores novel therapeutic targets for both sporadic and familial AD. It would be better if they explain shortly how manage to explore novel therapotic targets.

Reviewer #2: I uploaded my comments as word file.

As it is written in the manuscript, the sample size and statistical analysis seem appropriate. The authors have made some data available as supporting material, including RT-PCR results, and ethics and animal documents. However, these documents are all in Chinese, so I cannot check them. I did not see any uncropped blot membranes, which could be a concern. As I am not a native English speaker, I cannot evaluate the language of the manuscript.

Reviewer #3: The study titled "Stem cell factor restrains endoplasmic reticulum stress-associated apoptosis through c-kit receptor activation of JAK2/STAT3 axis in hippocampal neuronal cells" is a well-designed article that clearly explains the purpose and study methods to the reader-scientists. The work can be accepted without requiring revision. However, it is recommended that authors make a few simple suggestions at the proof-reading stage, if appropriate for the Editor.

The author should add introduction part how to manage these inhibitors as AD theraputics and emphasis its importance more.

Please correct in Material and Methods part test names fully for instance “3-(4,5)-dimethylthiahiazo (-z-y1)-3,5-di- phenytetrazoliumromide (MTT) Test” or “5-Bromo-2'-deoxyuridine (BrdU) Test”

Figures microscope photos does not look clear, a clearer image can be added for all figures

6. PLOS authors have the option to publish the peer review history of their article (what does this mean?). If published, this will include your full peer review and any attached files.

Reviewer #1: **Yes: **Prof. G. OZHAN

Reviewer #2: No

Reviewer #3: No

---

## [Author Response · Author response to Decision Letter 0]

25 Aug 2024

We extend our profound appreciation to the editor expert for their meticulous and outstanding efforts. We are dedicated to adhering to the editor's guidance and make improvements to the manuscript to enhance its quality.

---

## [Editor Report · Decision Letter 1]

9 Sep 2024

Stem cell factor restrains endoplasmic reticulum stress-associated apoptosis through c-kit receptor activation of JAK2/STAT3 axis in hippocampal neuronal cells

PONE-D-24-18303R1

Dear Dr. SHEN,

We’re pleased to inform you that your manuscript has been judged scientifically suitable for publication and will be formally accepted for publication once it meets all outstanding technical requirements.

Kind regards,

Abhishek Kumar Singh, Ph.D.

Academic Editor

PLOS ONE

---

## [Editor Report · Acceptance letter]

6 Nov 2024

PONE-D-24-18303R1 

PLOS ONE

Dear Dr. SHEN, 

I'm pleased to inform you that your manuscript has been deemed suitable for publication in PLOS ONE. Congratulations! Your manuscript is now being handed over to our production team.

Kind regards, 

on behalf of

Dr. Abhishek Kumar Singh 

Academic Editor

PLOS ONE